# Flexible and Highly Sensitive Humidity Sensor Based on Sandwich-Like Ag/Fe_3_O_4_ Nanowires Composite for Multiple Dynamic Monitoring

**DOI:** 10.3390/nano9101399

**Published:** 2019-10-01

**Authors:** Maojiang Zhang, Minglei Wang, Mingxing Zhang, Long Qiu, Yinjie Liu, Wenli Zhang, Yumei Zhang, Jiangtao Hu, Guozhong Wu

**Affiliations:** 1CAS Center for Excellence on TMSR Energy System, Shanghai Institute of Applied Physics, Chinese Academy of Sciences, No. 2019 Jialuo Road, Jiading District, Shanghai 201800, China; zhangmaojiang@sinap.ac.cn (M.Z.); wangminglei@sinap.ac.cn (M.W.); zhangmingxing@sinap.ac.cn (M.Z.); qiulong@sinap.ac.cn (L.Q.); liuyinjie@sinap.ac.cn (Y.L.); zhangwenli@sinap.ac.cn (W.Z.); 2School of Nuclear Science and Technology, University of Chinese Academy of Sciences, Beijing 100049, China; 3School of Physical Science and Technology, Shanghai Tech University, Shanghai 200031, China; 4State Key Laboratory for Modification of Chemical Fibers and Polymer Materials, Donghua University, Shanghai 201620, China; zhangym@dhu.edu.cn

**Keywords:** Fe_3_O_4_ nanowire, silver, humidity sensor, respiration monitoring

## Abstract

Functional textiles with unique functions, including free cutting, embroidery and changeable shape, will be attractive for smart wear of human beings. Herein, we fabricated a sandwich-like humidity sensor made from silver coated one-dimensional magnetite nanowire (Fe_3_O_4_ NW) arrays which were in situ grown on the surface of modified polypropylene nonwoven fabric via simultaneous radiation induced graft polymerization and co-precipitation. The humidity sensor exhibits an obvious response to the relative humidity (RH) ranging from RH 11% to RH 95% and its response value reaches a maximum of 6600% (*ΔI/I_0_*) at 95% relative humidity (RH). The humidity sensor can be tailored into various shapes and embroidered on its surface without affecting its functionalities. More interesting, the intensity of its response is proportional to the size of the material. These features permit the sensor to be integrated into commercial textiles or a gas mask to accurately monitor a variety of important human activities including respiration, blowing, speaking and perspiration. Moreover, it also can distinguish different human physical conditions by recognizing respiration response patterns. The sandwich-like sensor can be readily integrated with textiles to fabricate promising smart electronics for human healthcare.

## 1. Introduction

Flexible electronics have received great attention due to their portability, miniaturization and potential applications to the internet of things and to human-machine interfaces. Wearable multifunctional electronics are particularly attractive because they permit the wearer to receive valuable and timely information about health without compromising the functionality or comfort of their garments. In recent decades, wearable sensors have developed substantially, with most reports focusing on the health monitoring via the perception of physiological activities such as blood pressure [1], pulse rate [2], oxygenation of the blood [3], skin temperature [4], brain activity [5], bodily motion [6] and respiration rate [7] to provide an insight into the user’s health. In addition, monitoring of the gas and water molecules emanating from the human body or from the environment is also considered to play an important role in personalized healthcare and medicine [8]. Thus, fabrication of humidity sensors that are flexible, wearable and ergonomic, for detecting and real time monitoring of the humidity of the human skin, breath and of the environment surrounding the human body, is necessary because it can quickly convert moisture levels into visual electrical signals, which greatly facilitates people’s lives. For example, real-time monitoring of moisture levels of human skin can provide a variety of important information about physiological, metabolic and health which can further aid on the evaluation of human health or of the effectiveness of cosmetics [9]. Moreover, the healing of skin wounds can also be assessed by monitoring the varieties of humidity levels around the wound [10]. The most attractive thing is that the wearable humidity sensor can be used to monitor respiration, which is one of the most important physiological signals in Pressure Saturation standard (BTPS) system because it can assist doctors in their diagnostics. Furthermore, a humidity sensor can realize non-contact monitoring, which reduces the risk of bacterial transmission and cross-infection as occurs in traditional contact monitoring [11].

So far, a bulk of materials are used for fabrication of humidity sensors such as conjugated polymers [12], sulphides [13], carbon-based nanomaterials [14], metal oxides [15], nanohybrids [16] and so on. Among them, one-dimensional (1D) semiconductor nanostructure are of particular interest because of extremely large surface-to-volume ratios and is considered to be the most suitable material platform for a good gas-sensing performance due to the excellent accessibility of target gases [17]. Many humidity sensors based on 1D semiconductor nanostructure have been successfully obtained. However, humidity sensors based on 1D nanomaterials are generally fabricated by adhesion or blending, which generally undergoes agglomeration or completely/partly shielded by other components. Furthermore, the application of the sensors is constrained due to a rigid structure or high humidity in which short circuit occurred when humidity exceeds a certain value. There remains the need to realize such flexible sensors and customize their sensing layer and electrode in the shape and size, in order to create a universal fabric that could be simply cut and stitched to different flexible surfaces. Finally, a high cost and complicated integration process must also be considered. Therefore, the realization of a humidity sensor with flexibility, wide response range and easy integration with other flexible electronic devices is vital for its practical application.

In situ formation of 1D semiconductors nanowires (NWs) on polymer supports offer advantages including flexibility, high sensitivity, low cost, minimal calibration requirement and the possibility of depositing on various kinds of matrices [18]. As far as we know, in situ formation of NWs on the surface of low heat-resistant polymer matrix is limited to the formation of ZnO NWs with a wurtzite crystal structure; however, the performance of these sensors was constrained by their bending characteristics because of the poor adhesion between organic supports and sensing layer [19]. Therefore, good adhesion between the organic supports and sensing layer is essential for the practical application of flexible sensor. In addition, due to the intrinsically limited sensitivity of ZnO NWs, it is difficult to perceive minor variation of relative humidity (RH); more efforts are needed. 

Recently, the electronic characteristics of 1D Fe_3_O_4_ NWs have attracted much attention due to their unique electron-transport mechanisms [20,21]. The Fe_3_O_4_ is known as a half-metal—the electrons can hop quickly among Fe^2+^ and Fe^3+^ ions on octahedral sites at room temperature, resulting in good electric conductivity [22] and high sensitivity to the electrical disturbances from water or other gaseous molecules. In our previous studies, we report a simple, reproducible and economical method for the in situ formation of Fe_3_O_4_ NW arrays on the surface of polypropylene nonwoven fabric (PP NWF). Further studies found that the Fe_3_O_4_ NW arrays obtained on the surface of PP NWF exhibited a good saturation magnetization and super sensitivity to external stimuli such as UV and moisture but the problem that Fe_3_O_4_ is easily oxidized by oxygen needs to be solved.

Herein, we prepared a high-performance, fabric-based flexible sandwich-like electronic sensor for real-time monitoring the varieties of humidity environment of the human body. Based on our previous study, highly oriented/durable Fe_3_O_4_ NW arrays were in situ formed on the flexible PP NWF by the radiation induced graft polymerization and surface coordination induced growth [23] (coded as PP-*g*-PAO/Fe_3_O_4_), subsequently silver nanoparticles (Ag NPs) were in situ formed on the surface of Fe_3_O_4_ NWs (coded as Ag@Fe_3_O_4_-MS, MS: multifunctional sensor). This structure not only reduces the oxidation of Fe_3_O_4_ NWs by oxygen in air but also enhances the device performance via doping to Fe_3_O_4_ NWs. The humidity sensor exhibits an obvious response to the relative humidity (RH) ranging from RH 11% to RH 95% and its response value reaches a maximum of 6600% (*ΔI/I_0_*) at 95% relative humidity (RH). This multifunctional Ag@Fe_3_O_4_-MS sensor also shows potential in monitoring other health-related parameters, including respiration monitoring, breathing humidity and skin humidity, based on its excellent humidity-sensing properties. The humidity sensor reported herein can be tailored into various shapes, surface embroidery and integrated into other textiles or already existing flexible electronic systems, thus it is a promising functional material for applications in flexible fabric-based wearable electronic products.

## 2. Experimental Section

### 2.1. Raw Materials 

Polypropylene nonwoven fabrics (PP NWFs) were procured from Haining Laisheng Equipment Co., Ltd. (Haining, China). Acrylonitrile (AN, AR), dimethyl sulfoxide (DMSO, AR), dimethylformamide (DMF, AR), sodium carbonate (Na_2_CO_3_, AR), hydroxylamine hydrochloride (NH_2_OH·HCl, AR), ferrous sulfate heptahydrate (FeSO_4_·6H_2_O, CR), ferric chloride hexahydrate (FeCl_3_·6H_2_O, CR), ammonia solution (NH_4_OH, 25–28%), glucose, silver nitrate (AgNO_3_, AR), tartaric acid (AR), sodium hydroxide (NaOH, AR) and ethanol (AR) were purchased from Sinopharm Chemical Reagent Co., Ltd. (Shanghai, China). All reagents were used as received.

### 2.2. Fabrication of Ag@Fe_3_O_4_-MS Humidity Sensor

In situ growth of Fe_3_O_4_ NW arrays on PP NWF: PP-*g*-PAO/Fe_3_O_4_ was obtained by the radiation induced graft polymerization and surface coordination induced growth according to our previous report (Appendix A) [23]. The Ag@Fe_3_O_4_-MS was prepared in a modified silver mirror reaction based on self-assembly Ag NPs on the surface of PP-*g*-PAO/Fe_3_O_4_. AgNO_3_ (3.5 g), which was dissolved in deionized water (60 mL), to which then the NH_3_·H_2_O solution was added until a fine brown precipitate disappeared. Subsequently, 2.5 wt% NaOH (100 mL) were first added to the previous solution (turn black) and then a NH_3_·H_2_O solution was added until the final mixture became completely transparent. The PP-g-PAO/Fe_3_O_4_ (0.5 g) was introduced into this mixture by continuous ultrasonication for 30 min. By last a mixed solution of deionized water (54.5 mL) and ethanol (5.5 mL) containing 3.6 g/L tartaric acid and 40.9 g/L glucose was dropwise added to the ultrasonicated mixture at 25 °C for 10 min. The obtained product was once again ultrasonically and cleaned for 15 min: then it was repeatedly washed with deionized water until neutral condition and then dried in an oven at 60 °C. The product was designated as Ag@Fe_3_O_4_-MS.

### 2.3. Characterizations 

The morphologies and elemental distributions of the samples were identified using a field emission scanning electron microscope (FESEM, Merlin Compact, Zeiss, Jena, Germany) with an energy dispersive spectrometer (EDS). The functional groups of the fabrics were characterized by a Bruker Tensor 207 Fourier transform infrared spectrometer (FTIR, Bruker (Beijing), Inc., Beijing, China) in the 400 to 4000 cm^−1^ range. The electrical response of the sensors was tested at 9 V using a Keithley 7510 digital multimeter (Tektronix, Inc., Beaverton, OR, USA). The humidity-sensing was tested by placing the sensor in different RH atmospheres. Various RH environments were produced using different saturated salt solutions as listed as follows: LiCl (11%), MgCl_2_ (33%), Mg(NO_3_)_2_ (54%), NaCl (75%), KCl (85%) and KNO_3_ (95%) at 20 °C [24].

## 3. Results and Discussion

### 3.1. Design and Characterization of Ag@Fe_3_O_4_-MS Humidity Sensors

The proposed fabrication processes and hierarchical structure of the PP-*g*-PAO/Fe_3_O_4_ is illustrated in Appendix A. Graft chains act as a shape controller and stabilizer in the synthesis of Fe_3_O_4_ NWs leading to the formation of 1-D nanostructures of Fe_3_O_4_ NWs under benign conditions. Briefly, graft chains are covalently connected to the PP NWF at the molecular level and then chelated with iron ions to introduce seeds of the semiconductor on the PP substrate, which finally initiate the subsequent growth of Fe_3_O_4_ NWs. The grafted polymer chains and Fe_3_O_4_ NWs formed flexible three-dimensional network structures, by means of N-O-Fe and Fe-N chemical bonds, which effectively improve the bond strength between the PP NWF and Fe_3_O_4_ NWs (Appendix A). In the preparation of the humidity sensor, PP-*g*-PAO/Fe_3_O_4_ was used as a substrate (Figure 1a illustrates the design scheme of the Ag@Fe_3_O_4_-MS sensor). The Ag sheets deposited in situ on the surface layer of PP-*g*-PAO/Fe_3_O_4_ serves as electrodes and Ag NPs doped the sensing layer to form a Schottky barrier in the host materials to improve the sensitivity. If the humidity sensor can be arbitrarily cut or surface embroidery, like an ordinary fabric, to functionalize commercially available clothes by sewing, this will greatly simplify the application of the humidity sensor in smart clothing. However, universal wearable sensors are incapable of being cut into different shapes and sewed because of traditional electrode structures (Figure 1b) [25,26]. In this work, the flexible sandwich-like Ag@Fe_3_O_4_-MS can be cut into different shapes and integrated into other flexible textiles without affecting their functionalities (Figure 1c,d). The universal electrode material, such as silver paste or copper foil, can be replaced because of the in situ deposition of Ag NPs on the both surfaces of PP-*g*-PAO/Fe_3_O_4_. Figure 1e displays a cross-sectional SEM image and EDS mapping image of Ag@Fe_3_O_4_-MS. An important feature is that both the surface layers (two electrodes, about 100 μm) and subsurface layer (sensing layer, about 700 μm) of Ag@Fe_3_O_4_-MS form the sandwich structure (Figure 1e). Ag@Fe_3_O_4_-MS is designed by in-situ layer-by-layer assembly on the surface of PP NWF, which can effectively improve the bond strength between the Ag and Fe_3_O_4_. Ag@Fe_3_O_4_-MS realizes such flexible sensors and customizes their sensing layers and electrodes in the shape and size, in order to create a functional fabric that could be simply cut into promising wearable devices. Moreover, the Ag NPs are non-toxic and have an antibacterial effect, which is important when used in the monitoring processes like respiration and perspiration.

Figure 2a–d display the surface and subsurface SEM images of PP-*g*-PAO/Fe_3_O_4_ and Ag@Fe_3_O_4_-MS. Fe_3_O_4_ NW arrays covered onto the surface and subsurface of the PP-*g*-PAO after the radiation-induced graft polymerization and co-precipitation processes (Figure 2a,b). The linear shape of Fe_3_O_4_ NWs exhibit an average diameter of 31 nm and a typical length of 300 nm (length: diameter ratio ~10) [23]. As shown in Figure 2c,d, a fish scaly-like silver sheets only grew on the surface layer of PP-*g*-PAO/Fe_3_O_4_. The Ag sheets on the surface of Ag@Fe_3_O_4_-MS acts as self-assembly electrodes and it is crucial to simplify the signal transmission/reception and device structure. In the Ag@Fe_3_O_4_-MS, Fe_3_O_4_ is an electron-rich material that accelerates the reduction of silver ions to Ag sheets, so the Ag sheets cannot further diffuse into the inner pores. To further confirm the distribution and the existence of Ag, Fe, C, N and O elements, EDS mapping of the surface and subsurface layer of Ag@Fe_3_O_4_-MS was carried out and the results are exhibited in Appendix A. The EDS data clearly show that silver presents a gradient distribution from surface layer to subsurface layer, which can also be confirmed in Figure 1e. The signals of other elements such as Fe, C, N and O are partial or total shielded due to the coverage of silver and the limited detection depth of XPS technology [27]. By the in situ reduction of Ag^+^, Ag NPs aggregated into a dense conducting layer on the surface of PP-*g*-PAO/Fe_3_O_4_ and formed an Ag@Fe_3_O_4_ structure on the subsurface. As a result, the Ag@Fe_3_O_4_ structure strengthens the interfacial interaction between them.

To determine the varieties of chemical structures during the preparation process, FTIR measurements were carried out and the results are presented in Figure 2e,f. Compared with the spectrum of the trunk PP NWF, the spectrum of the grafted materials (denoted as PP-*g*-PAN) shows new absorption bands at 2243 cm^−1^, which is attributed to -CN stretching. Nevertheless, the intensity of this peak decreased after reaction with NH_2_OH indicating the conversion of the nitrile groups into the amidoximation groups. In Figure 2e, the new characteristic adsorption bands at 3200–3500, 1657 and 940 cm^−1^ are due to the adsorption of -OH, -C = N- and -N-O- of the AO groups, respectively. These results indicate that PAO groups are successfully introduced onto the surface of PP NWF, which indicates that the functional interface layer was successfully fabricated. A peak at 616 cm^−1^ in PP-*g*-PAO/Fe_3_O_4_ spectrum was observed due to Fe-O stretching mode of the octahedral and tetrahedral sites (Figure 2f). Because of the formation of chemical bonds, electrons will transfer from the amidoxime group to Fe_3_O_4_ NWs, which increases the density of free electrons in Fe_3_O_4_ NWs. Compared with that in pure Fe_3_O_4_ (located at 585 cm^−1^) [28], the stretching bands of Fe-O moves toward a higher position. After the Ag NP layer was deposited in situ on the surface of the sample (PP-*g*-PAO/Fe_3_O_4_), the characteristic bands of Fe-O shifted to a lower position (587 cm^−1^) again. This data indicates that electrons migrate from amidoxime group to Fe_3_O_4_ NWs and then to silver. Ag@Fe_3_O_4_-MS retains the advantages of large specific surface area of nanowires to the greatest extent, which is conducive to the adsorption and desorption of water molecules. This special structure makes its dielectric properties more susceptible to moisture in the air, thereby increasing its sensitivity.

### 3.2. Humidity Sensing Properties of Ag@Fe_3_O_4_-MS

Humidity sensors, which convert variations in the concentration of water vapor in air to electrical signals, have shown promise for humidity monitoring in agricultural, industrial, medical, meteorological applications and intelligent packaging [29]. Figure 3a shows the humidity measuring system which was designed to verify the humidity sensing properties of the Ag@Fe_3_O_4_-MS sensor. Before all the sensing tests, the valve 3 is open, while the valves 1 and 2 are closed. For the humidity sensing test, the valves 1 and 2 are open (while the valve 3 is closed) and atmospheres with different relative humidity (RH = 11–95%) values were generated by passing the carrier gas (dry nitrogen) through different saturated salt solutions. The varieties of electric current through the Ag@Fe_3_O_4_-MS (dimensions: 20 × 15 mm^2^) at room temperature in these atmospheres was recorded as the humidity response. The relative current variation is defined as *ΔI/I_0_ = (I − I_0_)/I_0_*, where *I_0_* and *I* are initial current (in dry nitrogen) and changed current, respectively. The real-time dynamic response curve of Figure 3b shows that when the RH increased, the relative current showed a synchronous dramatic increase. Figure 3c plots the humidity sensing response as log10 (*I_RH_/I_Dry_*), where I_RH_ is the current at the tested humidity and *I_Dry_* is that at the lowest humidity (i.e., dry nitrogen). The obtained sensitivities are 2.14 at RH = 11% and 64.67 at RH = 95%. These data show that Ag@Fe_3_O_4_-MS has a positive response to the variations of humidity. The cycle performance is another significant factor in evaluating a humidity sensor. The present sensor exhibited high repeatability and stable response, with the peak intensity stronger than those of reported sensors based on other nanomaterials such as graphene oxide [30]. In the present study, the response and recovery times are defined by the sensor achieving 95% of the total current variation in the case of adsorption and desorption, respectively. The obtained response and recovery times of Ag@Fe_3_O_4_-MS are shown in Figure 3d. The response time increased with increasing RH but the recovery time showed an opposite trend, which will be discussed below in the Mechanism section. In the actual application process, oxygen may also affect the sensor’s performance, thus the effects of oxygen must be considered. Appendix A shows that oxygen has little influence to sensitivity of the humidity sensor and the effect of oxygen can be eliminated.

Flexible sensors, especially those used in smart apparel, also need to show consistent performance when fabricated in different dimensions. Figure 4 exhibits the curves of relative current versus the dimension of Ag@Fe_3_O_4_-MS humidity sensor and the specific size changes are also listed. It is interesting to find that, as the area of Ag@Fe_3_O_4_-MS decreases (corresponding dimensional changes are shown in Figure 4a), the relative current intensity is also reduced (Figure 4b). As shown in Appendix A, the Ag@Fe_3_O_4_-MS exhibits a linear response to variations of its relative area. This trend may be ascribed to the sandwich structure of Ag@Fe_3_O_4_-MS in terms of the migration of charge carriers among the metal, semiconductor and polymer insulators. As the electrodes are located on opposite sides of the sandwich structure, the current can only flow from one side to the other side of the sensor. When decreasing the area of the sensor, the number of conductive channels between the upper and lower electrodes is also reduced, resulting in a decrease in the current. Controlling the conductivity of a material via changing its area will play an important role in the fabrication of smart devices.

The sensing mechanism of Ag@Fe_3_O_4_-MS sensor is schematically represented in Figure 5. At low RH level, water molecules are chemisorbed on the material surface. As the humidity increases, physisorbed layers will be formed on top of the chemisorbed layer. At low RH, water molecules cover the surface discontinuously to hinder the charge transfer and the Ag@Fe_3_O_4_-MS exhibits a high resistance. However, after the attachment of silver onto the Fe_3_O_4_ NWs, electrons migrate from Fe_3_O_4_ NWs to the silver because of a difference in potential energy [31]. Simultaneously, a Schottky barrier will be formed at the junction, which improves the charge segregation and prevents the charge recombination. Thus, the deposited silver on Ag@Fe_3_O_4_-MS possess a high local charge density and stronger electrostatic field, which facilitates the dissociation of water molecules to produce H_3_O^+^ and OH^−^. The protons carry the charge through proton hopping in the well-known Grotthuss mechanism [32,33] to decrease the resistance of Ag@Fe_3_O_4_-MS. A longer response time and a shorter recovery time at higher RH levels were observed and attributed to capillary condensation—the pre-condensation nucleation increased the response time, whereas in the recovery period the liquid phase on the surface of Ag@Fe_3_O_4_-MS evaporated quickly due to the larger exposed area, which facilitates the escape of condensed water from the sensor surface. At a higher RH level, more water molecules are adsorbed on Ag@Fe_3_O_4_-MS due to the fish scaly-like silver sheets, silver NPs and the Fe_3_O_4_ NWs that possess a large aspect ratio and high surface-to-volume ratio. The large aspect ratio can contribute to the rapid transfer of water molecules from or to the interacting region. At the same time, it also increases the rate of charge carriers transversely crossing the barriers induced by molecular recognition along the Fe_3_O_4_ NWs [34]. Therefore, the resistance of the sensor device is further decreased. At last, the amidoxime (AO)-based graft chains can also help improve the conductivity of Ag@Fe_3_O_4_-MS. During the in situ formation of Fe_3_O_4_ NWs on the surface of PP NWF, the electron-rich amidoxime groups tend to bond to the facets of Fe_3_O_4_ NWs that have lower electron density. Then, electrons in the amidoxime groups are easily transferred to Fe_3_O_4_ NWs and eventually reach the fish scaly-like silver sheets, thereby increasing the local charge density and the electrostatic field strength. This process led to the formation of a high density of localized states (charge traps). After binding the charges during charge transport, these traps will polarize the surrounding molecules, which will, conversely, become more conductive and accelerate the charge accumulation and then polarize even more molecules. This positive feedback effect will improve the conductivity of Ag@Fe_3_O_4_-MS. As the humidity further increases, more water molecules are adsorbed on the surface of Ag@Fe_3_O_4_-MS, leading to capillary condensation. The hydrophilic and electron-rich amidoxime groups can easily form complexes with the water molecules and simultaneously a proton is transferred to water following the reaction of -NH_2_ + H_2_O → -NH^−^ + H_3_O^+^. As a result, the amidoxime (AO)-based graft chains would enhance the sensing response of Ag@Fe_3_O_4_-MS for RH variations.

Table 1 provided typical parameters of some previously reported humidity sensors based on nanowire. Ag/Fe_3_O_4_ NWs based humidity sensor can be cut into various shapes and integrated into other flexible textiles. In addition, Ag@Fe_3_O_4_-MS can work over a wide RH range (11–95%).

### 3.3. Health Monitoring

To investigate the potential application of this sensor for respiratory monitoring, Ag@Fe_3_O_4_-MS was fixed inside a medical respirator (Figure 6a). As one of the most important physiological signs, respiration has become indispensable in medical monitoring especially for sleep apnea (temporary cessation of breathing during sleep). As shown in Figure 6b, in the corresponding data, there is a sudden long time decrease of current through this humidity sensor to indicate the apnea process. To further demonstrate its breath sensing function (Figure 6b’), the breath data are processed through dashed lines of different colors (apnea: red, inhale: orange, exhale: green). This simulated apnea time is 16.6 s, which can be quickly detected by its slope (about 1 s). The Figure 6b’ also shows that the slope of the inhaled or exhaled breath is same at different times, respectively. These results can facilitate the machine to quickly identify breathing behavior through the change of its slope, recording the time of each process and alerting to sudden situations. In addition, the Ag@Fe_3_O_4_-MS was tested for three modes of breathing: normal, fast and deep breathing (Figure 6d). The results indicate that the output signals accurately reflect the rate and mode of respiration. The variation of peak current intensity may be ascribed to the adsorption of water molecules on the surface of Ag@Fe_3_O_4_-MS, when the exhaled air contains much moisture. During the in-breath, water molecules desorb from the surface due to evaporation, leading to a gradual decay of the current (Figure 6e). The response of the sensors is synchronous with the breathing, due to the special structure of the Ag@Fe_3_O_4_-MS. In addition, when the respiratory rate increased, the intensity of the output electrical signals also exhibited a downward trend. A possible reason for this phenomenon may be that faster breathing means more heat in the exhaled air, which leads to a higher sensor temperature and hindered moisture adsorption on the sensor surface. Interestingly, the Ag@Fe_3_O_4_-MS not only distinguished breathing through the mouth and nose (Figure 6c) but also can distinguish different words such as “hi” and “hello” (Appendix A). Two types of breathing have different slopes of moisture exhaled due to the different levels of moisture exhaled from the mouth and nose. Thus, the Ag@Fe_3_O_4_-MS exhibits excellent abilities in breath monitoring and distinguishing between different breathing processes.

For applications in smart apparel, it is highly desirable that the sensor can be cut into various shapes and embroidered on the garment surface. Due to its high flexibility, mechanical robustness and excellent sensitivity, Ag@Fe_3_O_4_-MS exactly satisfies these requirements (Figure 7a,d). Small pieces of Ag@Fe_3_O_4_-MS were sewn onto the surface of a face mask and embroidered onto the sleeve to monitor the effects of breathing and perspiration, respectively. The electrode placement plus the sandwich structure allows these sensors to be cut into different shapes and stitches without affecting the connection structure which can be constructed directly to replace copper foil or silver paste as electrodes. In personal healthcare, respiratory humidity can indicate a person’s hydration state. Figure 7b and c exhibit a relative current variation of Ag@Fe_3_O_4_-MS before and after drinking water. The volunteer abstained from water for four hours before the test, so the initial peak intensity was very low. After the volunteer drank water, the relative current firstly exhibited an enhancement and then a gradual decrease with time. The statistical data of the current variation for all measurements is shown in Figure 7c. Before drinking water, the current varied by about 278% while this value jumped to about 1330% after drinking water. Then, the variations of relative current were recorded every 20 min. Afterwards, the variations of relative current were 1185%, 885% and 603% at 20, 40 and 60 min, respectively, clearly showing water loss from the body. Compared with the air flow meter, the humidity sensor based on Ag@Fe_3_O_4_-MS not only monitors the respiratory mode and frequency but also provides early warning of dehydration in patients. Moreover, the sensor also has stable performance in long-term respiratory monitoring (Appendix A). Finally, the humidity sensor could also detect skin moisture when mounted on a bracelet or a sleeve (Figure 7e,f). When placed close to the skin, the relative current is enhanced, especially for people who sweats heavily after strenuous exercise (Figure 7f). Many other humidity sensors, especially those based on conductive materials (such as metal or graphene) can only detect low humidity and are prone to short circuit when encountering high humidity such as sweaty skin. In fact, we directly dripped water droplets onto the sensor surface and an enhanced relative current was observed that gradually returned to the original value in time and no short circuit appeared because of its sandwich structure (Appendix A) in which a PP insulation layer was present. To further demonstrate the reliability of Ag@Fe_3_O_4_-MS, NaCl solution (1 g/L) or milk were dropped onto the surface of sensor. As seen in Appendix A, Ag@Fe_3_O_4_-MS showed a significant conductance change upon a drop of NaCl solution (Ag@Fe_3_O_4_-MS/NaCl) and it exhibited a bigger varieties in current after 5 drops of NaCl solution. After that, they gradually returned to its original value in time. The relative current varieties of Ag@Fe_3_O_4_-MS demonstrated a similar regularity after dripping milk on the surface of the sensor (Ag@Fe_3_O_4_-MS/milk) but the peak intensity of Ag@Fe_3_O_4_-MS/milk is lower than that of Ag@Fe_3_O_4_-MS/NaCl. Ag@Fe_3_O_4_-MS was immersed into NaCl solution (0.1 g/L, simulated sweat) or milk (solid content ≥ 6.5%) for 10 min and then dried in an oven at 60 °C. The influence of NaCl solution on the humidity sensor can be eliminated (Appendix A), while after a long time of soaking in milk, the sensor lost its ability to respond to the variations in humidity (Appendix A). The cause behind this phenomenon ascribe to the coverage of butterfat. In all, the sandwich-like Ag@Fe_3_O_4_-MS not only exhibits high sensitivity and stability but also has the potential to cope with environmental changes.

## 4. Conclusions

Silver-doped Fe_3_O_4_ NW arrays are employed for the first time to fabricate a highly sensitive, robust, high-resolution, fast-response/recovery and wide detection range humidity sensor by integrating on a hydrophobic and flexible PP NWF substrate. The sensing layer was firmly attached on the organic substrate and exhibited good durability due to the formation of chemical bonds between the supports and sensing layer and mechanical rivet effect. The unique sandwich structure of the Ag@Fe_3_O_4_-MS contributes to the high sensitivity, low limit of detection down to 11% RH and a broad detection range (11–100% RH). Remarkably, Ag@Fe_3_O_4_-MS avoids the short-circuiting at high humidity that is a ubiquitous problem for other reported sensors. The Ag@Fe_3_O_4_-MS humidity sensor could be easily tailored into desired shapes or embroidered on the surface of the sensor using a commercial embroidery machine; more importantly, it can be integrated into commercial clothes to monitor various human activities such as breath, speaking, blow and noncontact sensation. Finally, due to the soft characteristic, portability and easy to tailor, the robustly flexible sensor can be applied in diverse smart sensing devices and multiparametric sensing platforms in future.

## Figures and Tables

**Figure 1 nanomaterials-09-01399-f001:**
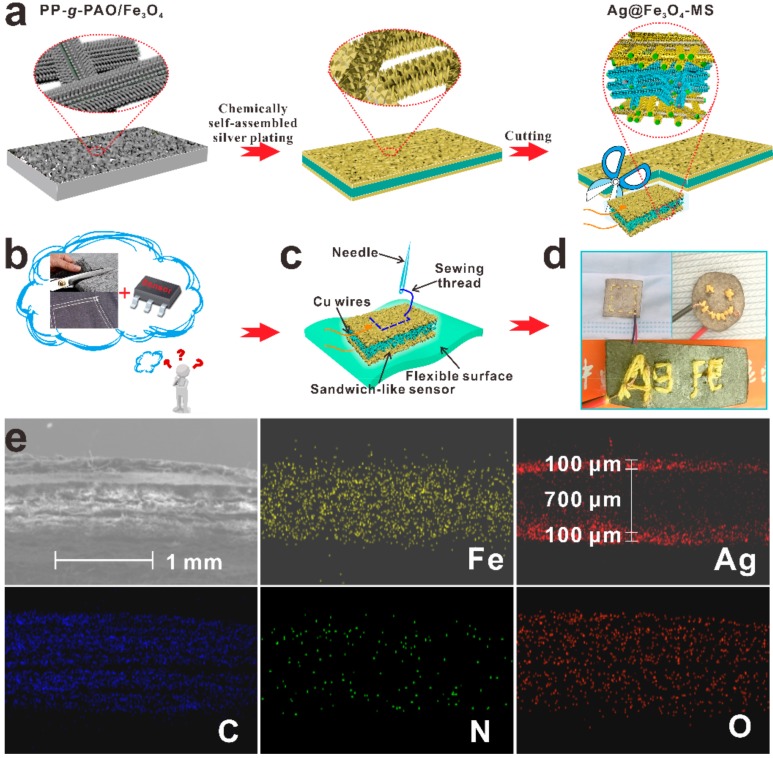
(**a**) Schematic fabrication process of the Ag@Fe_3_O_4_-MS sensor. (**b**) Ordinary fabric for cutting and sewing; a planar and strictly wafer-based sensor. (**c**) Illustration and (**d**) Photographs of humidity sensor: three-sublayered conductive architectures of the prepared Ag@Fe_3_O_4_-MS fabric sensor; e-fabric maintains the cutting and sewing properties of ordinary fabric. (**e**) Cross-sectional scanning electron microscopy (SEM) image and energy dispersive spectrometry (EDS) mapping image of Ag@Fe_3_O_4_-MS. Surface layer: 100 μm; subsurface layer: 700 μm.

**Figure 2 nanomaterials-09-01399-f002:**
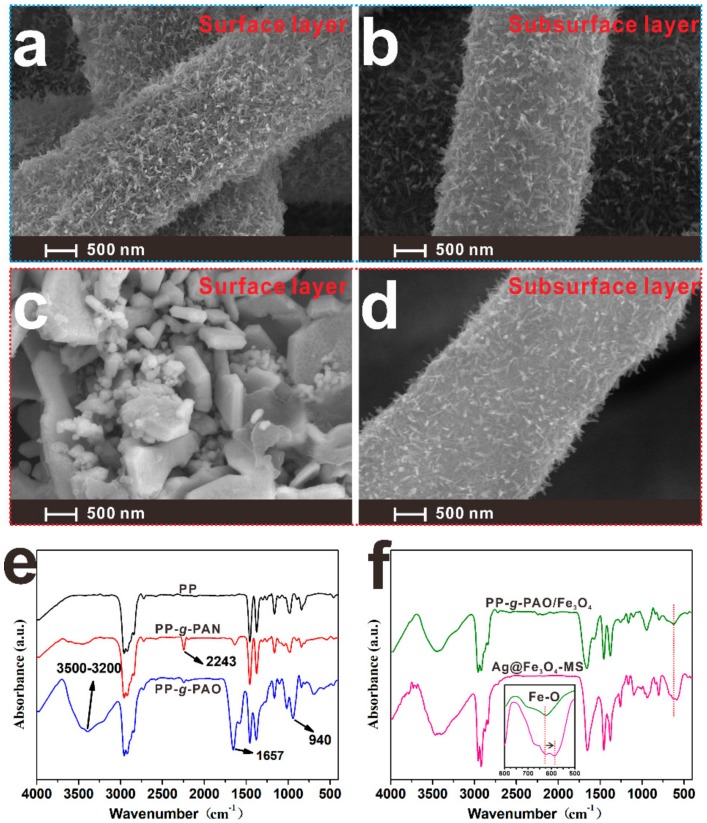
SEM images of (**a**,**b**) PP-*g*-PAO/Fe_3_O_4_ and (**c**,**d**) Ag@Fe_3_O_4_-MS. (**e**,**f**) Fourier transform infrared (FTIR) spectra of PP, PP-*g*-PAN, PP-*g*-PAO, PP-*g*-PAO/Fe_3_O_4_ and Ag@Fe_3_O_4_-MS.

**Figure 3 nanomaterials-09-01399-f003:**
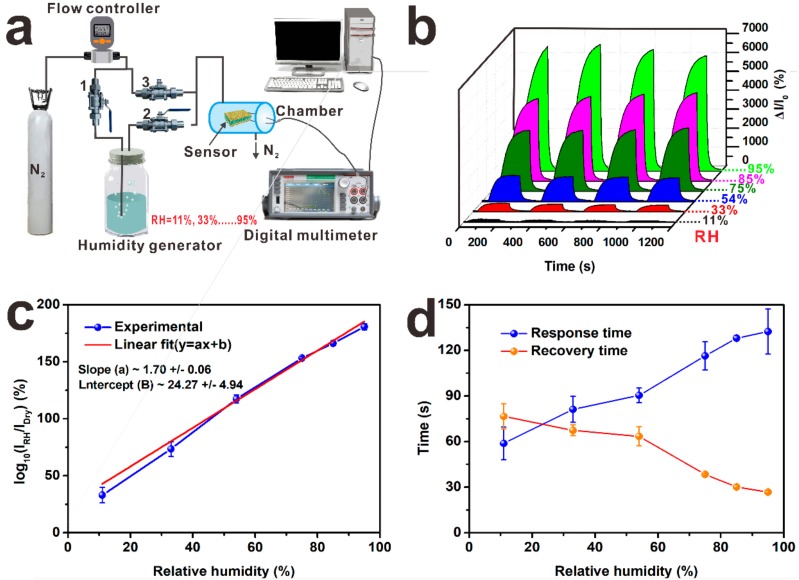
(**a**) Homemade humidity test system. (**b**) The repeatability characteristic of Ag@Fe_3_O_4_-MS at RH = 11–95%. The specimen dimensions are 20 mm length and 15 mm width. (**c**) Plot of log10 (*I_RH_/I_Dry_*) for Ag@Fe_3_O_4_-MS versus relative humidity (RH). (**d**) Responses of the sensor with respect to RH.

**Figure 4 nanomaterials-09-01399-f004:**
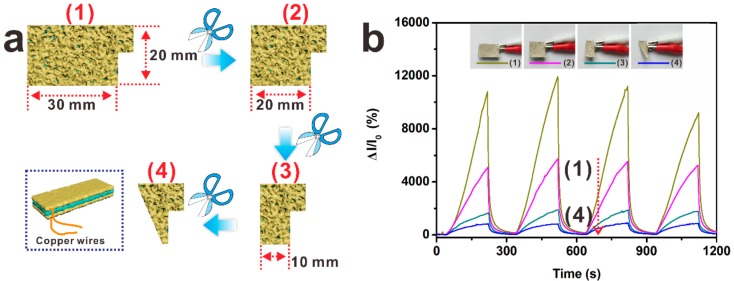
Free-cutting properties: (**a**) specific shapes and dimensions corresponding to (**b**). Insert: the electrodes of Ag@Fe_3_O_4_-MS can be constructed directly without silver paste and tailored to desired shapes. (**b**) Dependence of relative current on the dimensions and shapes of Ag@Fe_3_O_4_-MS at RH = 85%.

**Figure 5 nanomaterials-09-01399-f005:**
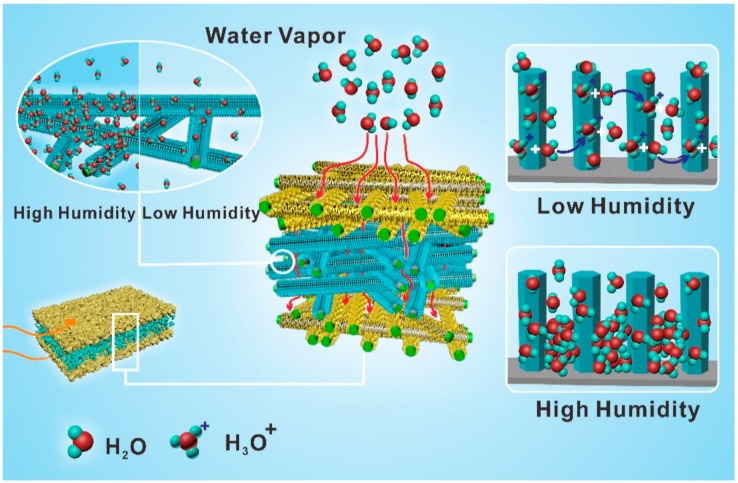
Schematic diagram of the sensing mechanism of Ag@Fe_3_O_4_-MS at high and low RH.

**Figure 6 nanomaterials-09-01399-f006:**
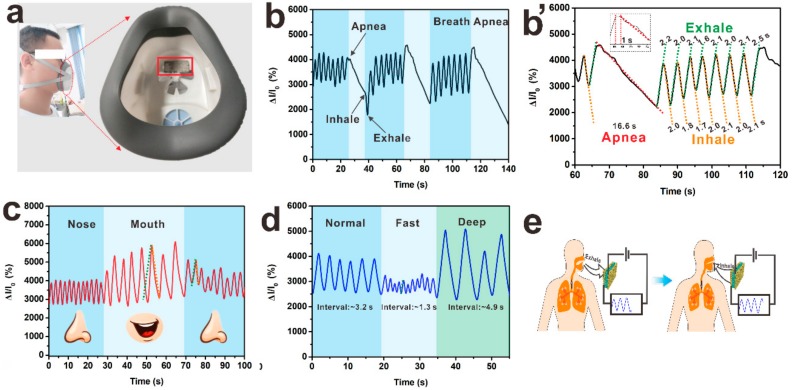
(**a**) Photograph of a human volunteer wearing a medical respirator with the humidity sensor fixed inside to monitor respiration. (**b**,**b’**) Variation in the relative current versus time during inhalation, exhalation and voluntary apnea. (**c**) Detection of the rate and strength of an adult breathing via the nose and mouth. (**d**) Signal variations for normal, fast and deep respiration. (**e**) The working mechanism of the Ag@Fe_3_O_4_-MS based sensor for detecting respiration.

**Figure 7 nanomaterials-09-01399-f007:**
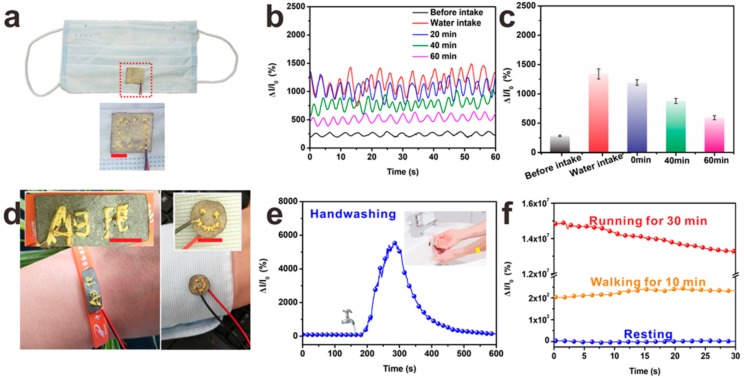
(**a**) Photograph showing the Ag@Fe_3_O_4_-MS fixed inside a common face mask. (**b**) Variation of the relative current through Ag@Fe_3_O_4_-MS in the breath before and after the volunteer drank water for a given time period afterwards. (**c**) Statistical results for the whole testing process of (**b**). (**d**) Tailoring and embroidery performances of Ag@Fe_3_O_4_-MS. (**e**,**f**) Performance in monitoring changes of humidity on the skin surface. Scale: 10 mm.

**Table 1 nanomaterials-09-01399-t001:** Comparison of the Typical Parameters of Different humidity sensors.

Substrate	Sensor Materials	Flexible or Rigid	Free-Cutting	Detection Range (% RH)	Reference
Si	SnO_2_ NWs	rigid	no	30–85	[35]
poly(ethylene terephthalate)	TiO_2_ NWs	flexible	no	20–90	[36]
SiO/Si	ZnO NWs	rigid	no	10–90	[37]
Polyurethane	Ag NWs	flexible	no	0–80	[38]
Polypropylene	Ag/Fe_3_O_4_ NWs	flexible	free cutting/embroidery	11–95	This work

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
