# Peer review of "Flexible and Highly Sensitive Humidity Sensor Based on Sandwich-Like Ag/Fe3O4 Nanowires Composite for Multiple Dynamic Monitoring"

_nanomaterials, 2019, doi:10.3390/nano9101399_

Round 1

Reviewer 1 Report

The paper is of interesting scientific quality: a new compound for water detection with high sensitivity is reported. The ideas and concepts presented in the article are novel. The paper is well organized: however, some text corrections and clarifications are needed.

The title and abstract are appropriate. The conclusions are clear but to be better supported, further data should be presented. Some Diagrams can be improved.

1) Suggested text revision in line 25 page 1, “These features permit the sensor to be integrated into commercial textiles or”.

2) Suggested text revision in line 34 page 1, “potential applications on internet of things and on human-machine interface.”.

3) Suggested text revision in line 36 page 1, “valuable and timely information about health without compromising the functionality”.

4) Suggested text revision in line 41 page 1, “monitoring of the gas and water molecules emanating from the human body or from the environment is”. The same for line 49 of page 2.

5) Suggested text revision in line 43 page 1, “fabrication of humidity sensors that are flexible, wearable and ergonomic, for detecting and real time monitoring of the humidity of the human skin, breath, and of the environment surrounding the human body,”

6) Suggested text revision in line 45 page 2, what do authors mean with varities of humidity?

7) Suggested text revision in line 46 page 2, “For example, real-time monitoring of moisture levels of human skin can provide a variety of important information about physiological, metabolic, and health which can further aid on the evaluation of human health or of the effectiveness of cosmetics [9].”

8) Suggested text revision in line 52 page 2, “Saturation standard (BTPS) system because it can assist doctors in their diagnostics. Furthermore, a humidity sensor can realize non-contact monitoring, which reduces the risk of bacterial transmission and cross-infection as occurs in traditional contact monitoring [11].”

9) Suggested text revision in line 97 page 3, “This multifunctional Ag@Fe3O4-MS sensor also shows potential in monitoring other health-related parameters, including respiration monitoring, breathing humidity and skin humidity, based on its excellent humidity-sensing properties.”

10) Suggested text revision in line 106 page 3, “Co., Ltd. Acrylonitrile (AN, AR), while dimethyl sulfoxide (DMSO, AR), dimethylformamide (DMF, AR), sodium carbonate (Na2CO3, AR), hydroxylamine hydrochloride (NH2OH·HCl, AR), ferrous sulfate heptahydrate (FeSO4·6H2O, CR), ferric chloride hexahydrate (FeCl3·6H2O, CR), ammonia solution (NH4OH, 25–28%), glucose, silver nitrate (AgNO3, AR), tartaric acid (AR), sodium hydroxide (NaOH, AR), and ethanol (AR) were purchased from Sinopharm Chemical Reagent Co., Ltd.: all reagents were used as received.”

11) Suggested text revision in line 113 page 3, “In situ growth of Fe3O4 NW arrays on PP NWF: PP-g-PAO/Fe3O4 was obtained by the radiation induced graft polymerization and surface coordination induced growth according to our previous report (Figure S1) [23]. The Ag@Fe3O4-MS was prepared in a modified silver mirror reaction based on self-assembly Ag NPs on the surface of PP-g-PAO/Fe3O4. AgNO3 (3.5 g), which was dissolved in deionized water (60 mL), to which then the NH3·H2O solution was added until a fine brown precipitate appeared: subsequently 2.5 wt% NaOH (100 mL) were firstly added to the previous solution (turn black) and, then a NH3·H2O solution was added until the final mixture became completely transparent. The PP-g-PAO/Fe3O4 (0.5 g) was introduced into this mixture by continuous ultrasonication for 30 min. By last a mixed solution of deionized water (54.5 ml) and ethanol (5.5 ml) containing 3.6 g/L tartaric acid and 40.9 g/L glucose was dropwise added to the ultrasonicated mixture at 25 oC for 10 min. The obtained product was once again ultrasonically and cleaned for 15 min: then it was repeatedly washed with deionized water until neutral condition, and then dried in an oven at 60 °C. The product was designated as Ag@Fe3O4-MS.”. Additionally, do authors mean instead of “until a fine brown precipitate appeared:”, “until the fine brown precipitate disappeared:”.

12) Suggested text revision in line 146 page 4, “iron ions to introduce seeds of the semiconductor on the PP substrate, which finally initiate the subsequent growth of Fe3O4 NWs. The formed grafted polymer chains and the Fe3O4 NWs flexible three-dimensional network structures, by means of N-O-Fe and Fe-N chemical bonds, which effectively improve the bond strength between the PP NWF and Fe3O4 NWs (Figure S1). In the preparation of  the humidity sensor PP-g-150 PAO/Fe3O4 was used as a substrate (Figure 1a illustrates the design scheme of the Ag@Fe3O4-MS sensor). The Ag sheets deposited in situ on the surface layer of PP-g-PAO/Fe3O4 serve as electrodes, and Ag NPs doped the sensing layer to form a Schottky barrier in the host materials to improve the sensitivity. If the humidity sensor can be arbitrarily cut or surface embroidery, like an ordinary fabric, to functionalize commercially available clothes by sewing, this will greatly simplify the application of the humidity sensor in smart clothing.”

13) Suggested text revision in line 176 page 5, “NWs exhibit an average diameter of 31 nm and a typical length of 300 nm (length: diameter ratio ~10) [23].”

14) Suggested text revision in line 190 page 6, “To determine the varieties of chemical structures during the preparation process, FTIR measurements were carried out, and the results are presented in Figure 2e and f. Compared with the spectrum of the trunk PP NWF, the spectrum of the grafted materials (denoted as PP-g-PAN) shows new absorption bands at 2243 cm−1, which is attributed to -CN stretching. Nevertheless, the intensity of this”

15) Suggested text revision in line 197 page 6, “groups are successfully introduced onto the surface of PP NWF, which indicates that the functional interface layer was successfully fabricated.”

16) Suggested text revision in line 201 page 6, “Fe3O4 NWs, what increases the density of free electrons in Fe3O4 NWs. Compared”

17) Suggested text revision in line 204 page 6, “This data indicates that the electrons migrate from amidoxime group to Fe3O4 NWs and then to silver.”

18) Clarification in line 217, “was examined using the homemade testing system shown in Figure 3a.”. Why use N2 instead of dry air in the homemade testing system: in real conditions humidity is measures in air which is composed of Azote and Oxygen, and Oxygen is known to also have a role in humidity sensors electrical behaviour. I also cannot really understand your experimental setup. Does the Azote pass through the different saturated solutions and then injected in the measuring chamber or not: because the presented scheme seems to transmit that after a mixture with dry Azote is done? How does it really work? Perhaps fig 3a should be redrawn. Additionally, in the inset of figure 3d replace reset by recovery.

19) Clarification: The legend of figure 4 should be changed in order to be clear and not contain symbol repetitions (letter b appears twice in the images and in the legend).

20) Suggested text revision in line 240 page 7, “Flexible sensors, especially those used in smart apparel, also need to show consistent performance when fabricated in different dimensions.”

21) Clarification in page 243 page 7, “It is interesting to find that, as the dimension decreases, the relative current intensity is also reduced.”. Which dimension? All? Do they vary proportionally? Clarify and describe them.

22) Suggested text revision in line 246 page 7, “As the electrodes are located on opposite sides of the sandwich structure, the current can only flow from one side to the other side of the sensor. When decreasing the area of the sensor, the number of conductive channels between the upper and lower electrodes is also reduced, resulting in a decrease in the current.”

23) Suggested text revision in line 253 page 8, “At low RH level, water molecules are chemisorbed on the material surface. As the humidity increases, 254 physiosorbed layers will be formed on top of the chemisorbed layer.”

24) Suggested text revision in line 262 page 8, “The protons carry the charge through proton hopping in the well-known Grotthuss mechanism [31, 32] to decrease the resistance of Ag@Fe3O4-MS, as humidity increases”

25) Suggested text revision in line 276 page 8, “Then, electrons in the amidoxime groups are easily transferred to Fe3O4 NWs and eventually reach the fish scaly-like silver sheets,”

26) Clarifications regarding section “3.3. Health Monitoring”: a single test case or subject is not sufficient to demonstrate and validate its usability in the diverse plotted situations: more tests should be performed and reported, otherwise no conclusions can be made. I understand that authors start by stating that they wanted to investigate its potential usage, but the section evolution, behaviour analysis and discussion/support develops in a way that goes behind evaluation, I believe that, either further data is reported or then the section should be rewritten. Perhaps some of the supplementary material should be added to this section, once its intention is to present a potential application and not proofed concept.

Author Response

September 14, 2019

Dear Reviewer:

We would like to express our gratitude to you for your critical reading of our manuscript and comments. The comments are very valuable for us to improve our paper. Based on these comments and suggestions, we have made careful modifications to the original manuscript. All changes made to the text are in clearly highlighted. We hope the new version of this manuscript will meet your Journal’s standard. Below you will find our point-by-point responses to the editor and reviewers’ comments/questions.

Thanks again!

Yours sincerely,

Guozhong Wu, PhD, Professor

Shanghai Institute of Applied Physics, CAS, China

***Response to Reviewers’ questions and comments ***

General questions and comments

Response reviewer #1

Thanks for your valuable suggestions and comments. We carefully read your opinions and summarized the following questions.

R1: Why use N2 instead of dry air in the homemade testing system: in real conditions humidity is measures in air which is composed of Azote and Oxygen, and.

The difference in humidity was measured using a common approach [ACS Appl. Mater. Interfaces 2019, 11, 24533−24543; Biosensors and Bioelectronics 2018, 116, 123–129], in which the humidity was controlled by flowing high purity nitrogen into different saturated salt solutions and the moisture was introduced to the sample chamber. In addition,the influence of oxygen atmosphere on the humidity sensor can be eliminated, see Figure S4.

R2: Does the Azote pass through the different saturated solutions and then injected in the measuring chamber or not: because the presented scheme seems to transmit that after a mixture with dry Azote is done? How does it really work? Perhaps Figure 3a should be redrawn.

Thanks for your advice, Figure 3a has been redrawn. In detail, “Before all the sensing tests, the valve 3 is open, while the valves 1 and 2 are closed. For the humidity sensing test, the valves 1 and 2 are open (while the valve 3 is closed), and  the atmospheres with different relative humidity (RH = 11–95%) values were generated by passing the carrier gas (dry nitrogen) through different saturated salt solutions. The variety of electric current through the Ag@Fe3O4-MS at room temperature in these atmospheres was recorded as the humidity response.”

R3: Additionally, in the inset of Figure 3d replace reset by recovery.

In the revised manuscript, this correction has been done.

R4: The legend of Figure 4 should be changed in order to be clear and not contain symbol repetitions (letter b appears twice in the images and in the legend).

To make them clearly, the legend of Figure 4 including (a), (b), (c) and (d) have been replaced by (1), (2), (3) and (4) in the revised manuscript, respectively.

R5: It is interesting to find that, as the dimension decreases, the relative current intensity is also reduced.” Which dimension? All? Do they vary proportionally? Clarify and describe them.

To clarify description, in the revised manuscript, “dimension” was replaced by “area”. The sentence was rewritten. “It is interesting to find that, as the area of Ag@Fe3O4-MS decreases (the dimension of Ag@Fe3O4-MS is shown in Figure 4a), the relative current intensity is also reduced (Figure 4b). As shown in Figure S5, the Ag@Fe3O4-MS exhibits a linear response to the variation of its relative area.”

Figure S5: Relative current intensity for Ag@Fe3O4-MS versus relative areas. (Relative area is defined as the ratio of sample area to minimum sample area.)

R6: Health Monitoring: a single test case or subject is not sufficient to demonstrate and validate its usability in the diverse plotted situations: more tests should be performed and reported, otherwise no conclusions can be made. I understand that authors start by stating that they wanted to investigate its potential usage, but the section evolution, behaviour analysis and discussion/support develops in a way that goes behind evaluation, I believe that, either further data is reported or then the section should be rewritten. Perhaps some of the supplementary material should be added to this section, once its intention is to present a potential application and not proofed concept.

As you mentioned, the purpose of this chapter is to explore the potential application of this sensing material. This paper focuses on the responsiveness of materials to external stimuli; therefore, only one volunteer was invited as a stimulus to observe the corresponding ability of materials to external stimuli. This study does not focus on medical research significance. In addition, we have made some necessary modifications in the revised manuscript, hoping to meet your requirements.

In detail, “To further demonstrate the reliability of Ag@Fe3O4-MS, NaCl solution (1 g/L) or milk were dropped onto the surface of sensor. As seen in Figure S9a, Ag@Fe3O4-MS showed a significant conductance change upon a drop of NaCl solution (Ag@Fe3O4-MS/NaCl), and it exhibited a bigger varieties in current after 5 drops of NaCl solution. After that, they gradually returned to its original value in time. The relative current varieties of Ag@Fe3O4-MS demonstrated a similar regularity after dripping milk on the surface of the sensor (Ag@Fe3O4-MS/milk), but the peak intensity of Ag@Fe3O4-MS/milk is lower than that of Ag@Fe3O4-MS/NaCl. Ag@Fe3O4-MS was immersed into NaCl solution (0.1 g/L, simulated sweat) or milk (solid content 6.5 %) for 10 min, and then dried in an oven at 60 °C. The influence of NaCl solution on the humidity sensor can be eliminated (Figure S9b), while after a long time of soaking in milk, the sensor lost its ability to respond to the varieties of humidity (Figure S9b). The cause behind this phenomenon ascribe to the coverage of butterfat. In all, the sandwich-like Ag@Fe3O4-MS not only exhibits high sensitivity and stability but also has the potential to cope with environmental changes.”

Figure S9. (a) Ag@Fe3O4-MS sensor responses to a NaCl solution ( or milk) droplet placed on it, and then five NaCl solution (or milk) droplets; NaCl solution is 1g/L. (b) Dependence of relative current of Ag@Fe3O4-MS at RH = 95%, before and after the sensor was assessed in additional NaCl solution or milk; insert of corresponding photographs. Scale: 10 mm.

Reviewer 2 Report

Present manuscript concerns the development of a sandwich-like humidity sensor based on Ag/Fe3O4 nanowires, in situ grown onto modified polypropylene nonwoven fabric surface. The system is characterized by laboratory tests and has been proposed for different application for human healthcare.

The work results interesting and well written. Sections related to design, fabrication and physical characterization of device result exhaustive explained. However, some aspects related to chemical characterization and sensor application for health monitoring require to deepen.

Main comments:

Please specify in the text the dimension of sensor used for humidity sensing tests, which results are shown in fig 3. It is reported that measurements have been performed using different saturated salt solutions (LiCl, MgCl2,Mg(NO3)2,KCl,NaCl and KNO3). Authors kindly justify this choice and explain if and how the salt influence the sensor performances.

Which salt solution has been used for measurements reported in fig. 3b?

Please specify also which kind of water was used for saturated salt solutions.

Authors reports that Ag/Fe3O4-MS has linear response to RH variation, but it is in disagree respect the plot reported in fig 3c where is represented log10(IRH/IDry) vs RH. I suggest to consider also this aspect. Author reported some results concerning the influence of area and shape of the device respect the sensor performance. Is it possible to define an equation or a relationship that describes this phenomenon? Concerning “Health monitoring” section, please specify the dimension of used sensors in different application. Please, explain better in the text how it is obtained fig 7b. At page 10 line 341 it is reported that the variation of relative current after drinking water has been monitored after 10, 30 and 60 min, while in fig 7b and c values are referred to 20, 40 and 60 min.

Details:

The sentence at page 7 line 216 “the humidity performance of our sensor” is not properly correct. Please rephrase that Pag 7 line 216, please put ref [44] in correct form.

Author Response

September 14, 2019

Dear Reviewer:

We would like to express our gratitude to you for your critical reading of our manuscript and comments. The comments are very valuable for us to improve our paper. Based on these comments and suggestions, we have made careful modifications to the original manuscript. All changes made to the text are in clearly highlighted. We hope the new version of this manuscript will meet your Journal’s standard. Below you will find our point-by-point responses to the editor and reviewers’ comments/questions.

Thanks again!

Yours sincerely,

Guozhong Wu, PhD, Professor

Shanghai Institute of Applied Physics, CAS, China

***Response to Reviewers’ questions and comments ***

General questions and comments

Response reviewer #2

R1: Please specify in the text the dimension of sensor used for humidity sensing tests, which results are shown.

The dimension of sensor used for humidity sensing tests in Figure 3 is 20×15 mm2, which has been added. In detail, “The varieties of electric current through the Ag@Fe3O4-MS (dimensions: 20×15 mm2) at room temperature in these atmospheres were recorded as the humidity response.”

R2: It is reported that measurements have been performed using different saturated salt solutions (LiCl, MgCl2, Mg(NO3)2, KCl, NaCl and KNO3). Authors kindly justify this choice and explain if and how the salt influence the sensor performances. Which salt solution has been used for measurements reported in Figure 3b? Please specify also which kind of water was used for saturated salt solutions.

According to the literature, the humidity was controlled using a common approach in which the high purity nitrogen flowed into different saturated salt solutions and carried the moisture to the sample chamber [ACS Appl. Mater. Interfaces 2019, 11, 24533−24543; Biosensors and Bioelectronics 2018, 116, 123–129]; moreover, the salts will not enter the reaction chamber during this process. Deionized water (18 MΩ cm) was used for all saturated salt solutions. The obtained saturated salt solution is used to measure the humidity response in experimental section (2.3). In detail, “Various RH environments were produced using different saturated salt solutions as listed as follows: LiCl (11%), MgCl2 (33%), Mg(NO3)2 (54%), NaCl (75%), KCl (85%), and KNO3 (95%) at 20 oC (in 2.3. characterization).”

R3: Authors reports that Ag/Fe3O4-MS has linear response to RH variation, but it is in disagree respect the plot reported in Figure 3c where is represented log10(IRH/IDry) vs RH. I suggest to consider also this aspect. Author reported some results concerning the influence of area and shape of the device respect the sensor performance. Is it possible to define an equation or a relationship that describes this phenomenon?

Thanks for your kind reminding. In the revised manuscript we replaced “linear” by “positive” to make the language accurate. “These data show that Ag@Fe3O4-MS has a positive response to the variations of humidity.”

Figure S5 shows that Ag@Fe3O4-MS has a linear response to its variations relative area. Equation: y=-1925.93×x-2007.44, R=0.97.

Figure S5: Relative current intensity for Ag@Fe3O4-MS versus relative areas. (Relative area is defined as the ratio of sample area to minimum sample area.)

R4: Concerning “Health monitoring” section, please specify the dimension of used sensors in different application. Please, explain better in the text how it is obtained Figure 7b. At page 10 line 341 it is reported that the variation of relative current after drinking water has been monitored after 10, 30 and 60 min, while in Figure 7b and c values are referred to 20, 40 and 60 min.

The dimension of used sensors in different application has been added in Figure 7a and d.

In the revised manuscript, we have carefully check original records to ensure the values in graphic Figure are corresponding to the text. The volunteer abstained from water for four hours before the test, and the relative currents were measured before and after the volunteer drank water. Then, the variations of relative current were recorded every 20 min (20, 40, 60 min).

Figure 7. (a) Photograph showing the Ag@Fe3O4-MS fixed inside a common face mask. (b) Variation of the relative current through Ag@Fe3O4-MS in the breath before and after the volunteer drank water for a given time period afterwards. (c) Statistical results for the whole testing process of (b). (d) Tailoring and embroidery performances of Ag@Fe3O4-MS. (e, f) Performance in monitoring changes of humidity on the skin surface. Scale: 10 mm.

R5: The sentence at page 7 line 216 “the humidity performance of our sensor” is not properly correct. Please rephrase that Pag 7 line 216, please put ref [44] in correct form.

The sentence at page 7 line 216 “The humidity performance of our sensor was examined using the homemade testing system shown in Figure 3a.” was replaced by “Figure 3a shows the humidity measuring system which was developed to verify the humidity sensing properties of the Ag@Fe3O4-MS sensor.”

We corrected ref [44] in correct form.

Reviewer 3 Report

The manuscript reports the design, fabrication, and testing of a fabric-based flexible sandwich-like electronic sensor for real-time monitoring of humidity.

English language editing and revisions should be carried out. There are significant syntax and grammar issues.

The authors should consider a more significant comparison of the presented sensors to other nanowire humidity sensors. A table for comparison would be much appreciated and of great value to the report.

The authors claim utility for wearable and textile based sensing. However no reliability or fouling studies are conducted. At a bare minimum, the sensor should be assessed in additional biological media, salt impact, any other representative fluids. Do the nanowires foul? Analysis of the materials after operation should be included. 

Author Response

September 14, 2019

Dear Reviewer:

We would like to express our gratitude to you for your critical reading of our manuscript and comments. The comments are very valuable for us to improve our paper. Based on these comments and suggestions, we have made careful modifications to the original manuscript. All changes made to the text are in clearly highlighted. We hope the new version of this manuscript will meet your Journal’s standard. Below you will find our point-by-point responses to the editor and reviewers’ comments/questions.

Thanks again!

Yours sincerely,

Guozhong Wu, PhD, Professor

Shanghai Institute of Applied Physics, CAS, China

***Response to Reviewers’ questions and comments ***

General questions and comments

Response reviewer #3

R1: English language editing and revisions should be carried out. There are significant syntax and grammar issues.

Thanks for your kind reminding, the manuscript has been carefully revised.

R2: The authors should consider a more significant comparison of the presented sensors to other nanowire humidity sensors. A table for comparison would be much appreciated and of great value to the report.

Table 1 provided typical parameters of some previously reported humidity sensors based on nanowire. Ag/Fe3O4 NWs based humidity sensor can be cut into various shapes, and integrated into other flexible textiles. In addition, Ag@Fe3O4-MS can work over a wide RH range (11–95%).

Table 1. Comparison of the Typical Parameters of Different humidity sensors

Substrate

Sensor materials

flexible or rigid

Free-cutting

detection range (% RH)

Reference

Si

SnO2 NWs

rigid

no

30-85

[1]

poly(ethylene terephthalate)

TiO2 NWs

flexible

no

20-90

[2]

SiO/Si

ZnO NWs

rigid

no

10-90

[3]

Polyurethane

Ag NWs

flexible

no

0-80

[4]

Polypropylene

Ag /

Fe3O4 NWs

flexible

free cutting/

embroidery

11-95

This work

R3: The authors claim utility for wearable and textile based sensing. However no reliability or fouling studies are conducted. At a bare minimum, the sensor should be assessed in additional biological media, salt impact, any other representative fluids. Do the nanowires foul? Analysis of the materials after operation should be included.

More works related to the fouling studies have been added to the revised manuscript.

In detail, “To further demonstrate the reliability of Ag@Fe3O4-MS, NaCl solution (1 g/L) or milk were dropped onto the surface of sensor. As seen in Figure S9a, Ag@Fe3O4-MS showed a significant conductance change upon a drop of NaCl solution (Ag@Fe3O4-MS/NaCl), and it exhibited a bigger varieties in current after 5 drops of NaCl solution. After that, they gradually returned to its original value in time. The relative current varieties of Ag@Fe3O4-MS demonstrated a similar regularity after dripping milk on the surface of the sensor (Ag@Fe3O4-MS/milk), but the peak intensity of Ag@Fe3O4-MS/milk is lower than that of Ag@Fe3O4-MS/NaCl. Ag@Fe3O4-MS was immersed into NaCl solution (0.1 g/L, simulated sweat) or milk (solid content 6.5 %) for 10 min, and then dried in an oven at 60 °C. The influence of NaCl solution on the humidity sensor can be eliminated (Figure S9b), while after a long time of soaking in milk, the sensor lost its ability to respond to the varieties of humidity (Figure S9b). The cause behind this phenomenon ascribe to the coverage of butterfat. In all, the sandwich-like Ag@Fe3O4-MS not only exhibits high sensitivity and stability but also has the potential to cope with environmental changes.”

Figure S9. (a) Ag@Fe3O4-MS sensor responses to a NaCl solution ( or milk) droplet placed on it, and then five NaCl solution (or milk) droplets; NaCl solution is 1g/L. (b) Dependence of relative current of Ag@Fe3O4-MS at RH = 95%, before and after the sensor was assessed in additional NaCl solution or milk; insert of corresponding photographs. Scale: 10 mm.

Q. Kuang, C. Lao, Z. L. Wang, Z. Xie and L. Zheng, J. Am. Chem. Soc., 2007, 129, 6070-6071. D. Shen, M. Xiao, Y. Xiao, G. Zou, L. Hu, B. Zhao, L. Liu, W. W. Duley and Y. N. Zhou, ACS Applied Materials & Interfaces, 2019, 11, 14249-14255. S. Park, D. Lee, B. Kwak, H.-S. Lee, S. Lee and B. Yoo, Sensors Actuators B: Chem., 2018, 268, 293-298. L.-X. Liu, W. Chen, H.-B. Zhang, Q.-W. Wang, F. Guan and Z.-Z. Yu, Adv. Funct. Mater., 0, 1905197.

Round 2

Reviewer 1 Report

Some text corrections and clarifications are still needed.

further data should be presented.

All figures in supplementary material that are important should be in the paper

Reviewer 2 Report

No additional revisions are required